# Effect of Heating and Citric Acid on the Performance of Cellulose Nanocrystal Thin Films

**DOI:** 10.3390/polym15071698

**Published:** 2023-03-29

**Authors:** Emília Csiszár, Imola Herceg, Erika Fekete

**Affiliations:** 1Department of Physical Chemistry and Materials Science, Faculty of Chemical Technology and Biotechnology, Budapest University of Technology and Economics, Műegyetem rkp. 3., H-1111 Budapest, Hungary; hercegimola93@gmail.com (I.H.); bodine.fekete.erika@vbk.bme.hu (E.F.); 2Institute of Materials and Environmental Chemistry, Research Centre for Natural Sciences, Magyar Tudósok Körútja 2., H-1117 Budapest, Hungary

**Keywords:** cellulose nanocrystal (CNC), biocomposite film, plasticizer, citric acid, crosslinking, heat treatment, competitive interactions, morphology, water resistance, mechanical properties

## Abstract

Cellulose nanocrystals (CNCs) were extracted from bleached cotton by sulfuric acid hydrolysis. Thin films were prepared from the aqueous suspension of CNCs by casting and evaporation with 15% glycerol as a plasticizer. Our research aimed to create stable films resistant to water. The structure and the interactions of the films were modified by short (10 min) heating at different temperatures (100, 140, and 160 °C) and by adding different amounts of citric acid (0, 10, 20, and 30%). Various analytical methods were used to determine the structure, surface properties, and mechanical properties. The interaction of composite films with water and water vapor was also investigated. Heat treatment did not significantly affect the film properties. Citric acid, without heat treatment, acted as a plasticizer. It promoted the disintegration of films in water, increased water vapor sorption, and reduced tensile strength, resulting in flexible and easy-to-handle films. The combination of heat treatment and citric acid resulted in stable liquid-water-resistant films with excellent mechanical properties. A minimum heating temperature of 120 °C and a citric acid concentration of 20% were required to obtain a stable CNC film structure resistant to liquid water.

## 1. Introduction

Nowadays, research in the field of cellulose focuses on the production, testing, and exploration of possible applications of different nanocelluloses. Nanocelluloses are new materials with unique properties. They are biocompatible and biodegradable and can be produced from renewable natural materials. The three main groups of nanocelluloses are cellulose nanocrystals (CNC), nanofibrillated cellulose, and bacterial nanocellulose [1]. 

Of the three nanocelluloses, nanocrystalline cellulose is produced from different cellulose sources, most commonly by sulfuric acid hydrolysis. Sulfuric acid degrades the more readily available and less ordered amorphous parts of the cellulose, releasing the crystalline ordered parts. Cellulose nanocrystals are needle-like crystallites with lengths and diameters in the nanometer range. During sulfuric acid hydrolysis, negatively charged sulfate ester groups are formed on the surface of the crystals, and their repulsive effect inhibits the aggregation of the nanocrystals. As a result, the nanocrystalline cellulose whiskers form a stable aqueous suspension [2]. 

From the suspension, transparent films with excellent mechanical and optical properties can be cast, which can be used in various applications. During drying after film casting, hydrogen bonds are formed between the nanocrystals, which result in a stable structure with good barrier properties. However, the resulting structure can be disintegrated by water and broken down into nanocrystals [3]. The formation of covalent bonds between the nanocrystals, e.g., with amino-aldehyde resins [4], etc., can lead to a water-resistant, stable film structure, which is favorable for many applications. 

Chemical crosslinking of cellulose is a well-known reaction in the textile area and is widely used to improve the wrinkle recovery properties of cellulosic textiles. The traditional amino-aldehyde resins have recently been replaced with poly(carboxylic acids), such as 1,2,3,4-butanetetracarboxylic acid (BTCA) and citric acid (CA), in textile finishing. The reaction is based on the esterification of hydroxyl groups of cellulose, occurring at an elevated temperature between 165 and 175 °C for about 5 min [5,6,7]. 

Citric acid (CA) is a well-known, biodegradable, biocompatible, and natural poly(carboxylic acid) with three carboxylic acid groups and a hydroxyl group. The carboxylic acid groups can form ester bonds with the accessible hydroxyls of cellulose [8] and starch [9] in a heterogeneous or homogeneous reaction, respectively. The crucial step in the reaction is the formation of the anhydride of citric acid [10]. A thermochemical reaction occurs between CA and cellulose as CA requires high temperatures to condense to an anhydride by a dehydration reaction [11]. Subsequently, the highly reactive citric anhydride is involved in the ester formation, with cellulose creating a crosslinked structure, even with itself and other hydroxyl-containing compounds present. In the starch–glycerol system, CA can react not only with the hydroxyl groups of starch but also with hydroxyls of glycerol, added as a plasticizer into the system, forming an open network [9]. In the first reaction, the CA esterifies the glycerol, and then the glycerol ester reacts with the starch. It was also reported that CA as an additive can form strong hydrogen bond interactions with the starch molecule, improving the thermal and water stability but reducing the tensile strength of the starch-based products [12]. CA can act as a crosslinking agent and plasticizer to produce water-resistant starch-based packaging materials [13]. Combining CNC as a reinforcing agent and CA as a crosslinker in hemicellulose matrix resulted in films with improved physical properties (elastic modulus, elongation, water vapor barrier, and water resistance) and potential for various application areas in food packaging [14].

Although there are many publications on the properties of nanocrystalline cellulose films, only a few deal with improving their resistance to water. Research has shown that crosslinking with cyclic acid has significantly improved the water resistance of other cellulose-based materials, such as regenerated cellulose [15], and paper filter [10]. However, the ‘CNC-citric acid-plasticizer’ system in thin films and the effect of the crosslinking on film properties have hardly been investigated. Nanocrystalline cellulose is an excellent new nanomaterial. Citric acid is a natural, biocompatible, and biodegradable compound. Combining these two materials could lead to a significant expansion in the application area of nanocelluloses.

In this research, cellulose nanocrystals were extracted from bleached cotton by sulfuric acid hydrolysis, and thin, transparent, and colorless films were prepared from their aqueous suspension by casting and evaporation. In order to obtain stable CNC films resistant to liquid water, heating, citric acid addition, and their combination were applied. The effects were measured by varying the CA concentration and temperature. The interaction of CNC films with water vapor and liquid water was characterized by contact angle measurement, sinking, and swelling test. The results demonstrate that a stable film structure resistant to water can only be obtained by crosslinking. The crosslinking affects the surface properties and morphology of CNC films as well as their mechanical properties.

## 2. Materials and Methods

### 2.1. Materials 

Bleached cotton plain-weave fabric with an areal density of approximately 165 g/m^2^, provided by Pannon-Flax Linen Weaving Co., (Győr, Hungary), was used as cellulose source for CNC production. Even though the nanocellulose production technologies are primarily based on wood-based biomass [16], the selection of bleached cotton can be explained by its high (100%) cellulose content and good wettability. All chemicals were purchased from Sigma-Aldrich. Analytical-grade sulfuric acid (98%), citric acid (CA), glycerol (99.5%), and diiodomethane (99%) were also used for the experiments.

### 2.2. Production and Characterization of Cellulose Nanocrystals 

The nanocrystalline cellulose was prepared from 10 g of bleached cotton fabric, which was first ground to powder in a Retsch MM400 ball mill (Retsch GmBH, Haan, Germany). Sulfuric acid (64%) treatment at 45 °C for 25 min was used for the extraction of CNC. To stop the reaction, the CNC suspension was diluted with deionized water and allowed to settle in the refrigerator. Then, the CNC suspension was washed three times and centrifuged at 13,500 rpm at 5 °C for 10 min between washing steps on a Hermle Z326K refrigerated centrifuge (Labortechnik GmbH, Wehingen, Germany). For neutralizing, the suspension was dialyzed against tap water for five days. Dialysis was followed by ultrasonication performed at 60% amplitude for 10 min in an ice bath using a Sonics VCX-500 (Sonics & Materials, Inc., Newtown, CT, USA) ultrasonic immersion transducer. The suspension was then stored in a refrigerator. The final aqueous suspensions contained about 3% cellulose nanocrystals by weight [17]. The average yield was 37 ± 3%, calculated as a percentage of the initial weight of cotton powder.

Transmission electron microscopy (TEM) with a Morgagni 268D TEM (100 kV; W filament, top-entry; point-resolution = 0.5 nm, line-resolution = 0.3 nm) was used and the images at 40,000× magnifications were recorded with a Megaview III CCD camera (1376 × 1032 pixels). A drop of the diluted CNC suspension (0.05 w/w%) was deposited onto a copper grid covered by a thin carbon film and stained with 2% uranyl acetate. The average size of the cellulose nanoparticles and their aggregates present in the sonicated cotton-CNC suspension was measured by a Horiba Partica LA-950V2 laser diffraction particle size analyzer (LD-PSA). 

### 2.3. Casting of CNC Films 

CNC films were prepared from the aqueous suspension of cellulose nanocrystals by casting and evaporation. Since the CNC films are very brittle, glycerol as plasticizer was added to the CNC suspension in 15% on the CNC basis. The effect of citric acid as a crosslinker was investigated by adding citric acid to the suspensions at 10, 20, and 30% concentrations on a dry CNC basis. (Designation of films: CA-10%, CA-20%, CA-30%, respectively). The mixture was ultrasonicated at 60% amplitude for 30 s in an ice bath using an ultrasonic immersion transducer, which was introduced above (in Section 2.2). Rectangular films with a thickness of about 70 µm were cast from the aqueous CNC suspension on the surface of a polypropylene plastic sheet. After evaporating at room temperature for about 2 days, the films were easily detachable from the plastic sheet. The detached films were stored and tested in a conditioning room at 23 °C and 55% relative humidity (RH) [17]. 

The effect of heating on the performance of the neat films (containing only glycerol) and the composite films (containing glycerol and citric acid) was also investigated. A short 10 min long heat treatment was applied at 100, 120, and 160 °C in a drying oven (Venti-Line, VWR International Ltd., Leuven, Belgium). Then, the films were cooled down to ambient temperature and reconditioned. 

In some of the experiments, only selected compositions of CNC films were investigated because of the wide composition range studied in the research.

### 2.4. Characterization of CNC Films 

Morphology of the films was characterized by scanning electron microscopy (SEM) using JEOL JSM 6380 LA equipment (Jeol Ltd., Tokyo, Japan). SEM micrographs were taken of the fracture surface of films, which were frozen in liquid nitrogen and subsequently broken. 

Surface roughness of the CNC films was measured by a Dimension 3100 atomic force microscope (AFM) equipped with a Nanoscope IIIa controller (Digital Instruments/Veeco, Santa Barbara, CA, USA). A series of atomic force microscopy images in the scan size range 2–20 mm were acquired. These images are proper 3D profiles of the sample surface and can be represented in a so-called 3D view, where an axonometric colored picture is calculated from the measured spatial data representing a microscopic but perspective view of the surface. Raw measurement files were processed using the Nanoscope software by first applying 3rd order flattening. Root mean square roughness (R_q_) [18] was determined with the following Formula (1): (1)$$R_{q}=\sqrt{\frac{\sum_{i=1}^{n}z_{i}{}^{2}}{n}}$$
where $z_{i}$ represents the distance of the *i*th point from the mean plane.

Contact angles were measured at 23 °C and 55% relative humidity using a Rame-Hart contact angle goniometer (USA) with a camera and a drop image standard software of DT-Acquire. Liquid drops of 20 μL were deposited on each film and the image of drops was captured immediately by the camera. The values reported are the average of contact angles of at least 5 drops for each sample. To calculate the surface energy of the CNC films, contact angle measurement was carried out with two liquid probes: distilled water and diiodomethane. From the equilibrium contact angle data, the surface free energy was calculated by the Owens–Wendt Formula (2) [19]:(2)$$\gamma_{LV}\left(cos\theta +1\right)=2\;{\left(\gamma_{LV}^{d}\gamma_{SV}^{d}\right)}^{1/2}+2\;{\left(\gamma_{LV}^{p}\gamma_{SV}^{p}\right)}^{1/2}$$
where *γ_LV_*, *γ_LV_^d^*, and *γ_LV_^p^* are the surface tension of the liquid and that of its dispersion and polar components, respectively, used in the measurements. The values of *γ_LV_*, *γ_LV_^d^*, and *γ_LV_^p^* used for the calculations are 72.8, 21.8, and 51.0 mJ/m^2^ for distilled water and 51.0, 51.0, and 0 mJ/m^2^ for diiodomethane. *Γ_SV_^d^* and *γ_SV_^p^* are the dispersion and polar components of the surface free energy of films, respectively. The total surface free energy of the films was calculated by the following Equation (3): (3)$$\gamma_{S}^{total}=\gamma_{SV}^{d}+\gamma_{SV}^{p}$$

Water vapor sorption (WVS) measurement was performed on each of the films. Three parallel measurements of each were performed. First, the samples were placed in a desiccator at 55% relative humidity for 24 h. Then, the films were placed into Petri dishes and then kept in a Memmert-type air-circulation climate chamber (Memmert GmbH, Schwabach, Germany) with an RH of 80% at 23 °C. The mass of the samples was weighed at fixed intervals until the equilibrium water uptake was reached. The average length of the measurement was about two weeks.

For characterizing the disintegration of CNC films in water, a film sample (1 × 1 cm) was laid gently onto the surface of distilled water (50 mL) under orbital shaking at 100 rpm (Boeco OS 20, Hamburg, Germany) at room temperature and the elapsed time was recorded when the film started to disintegrate. 

The extent of swelling was characterized by the swelling rate (*SR*), which was calculated from the dry (*W_d_*) and swollen (*W_s_*) weight of a film. The latter was determined after immersion in water for 24 h. The excess water was removed from the sample’s surface and the mass was measured. The SR is defined as follows (4):(4)$$\mathrm{SR}\;\left(\%\right)=\frac{W_{\;s}-W_{d}}{W_{d}}\cdot 100$$

Mechanical properties were examined using an Instron 5566 tensile tester (Norwood, MA, USA) equipped with a 500 N load cell. At least ten specimens with the size of 7 × 50 mm were cut from each of the films in different series. They were tested at 10 mm/min crosshead speed and with 20 mm span length. 

## 3. Results

### 3.1. Cellulose Nanocrystals in Aqueous Suspension

As described in Section 2.2, cellulose nanocrystals were extracted from the cotton cellulose by sulfuric acid hydrolysis. Sulfuric acid degraded the more readily available and less ordered amorphous parts of cellulose, releasing the crystalline ordered parts (i.e., cellulose nanocrystals). The final product of the process is the aqueous suspension of cellulose nanocrystals. The suspension’s dry solid content (i.e., CNC content) is about 3%. The average yield was about 37 ± 3%, calculated as a percentage of the initial weight of cotton powder. In the CNC’s aqueous suspension, micron-sized particles can be detected by laser diffraction particle size analysis (Figure 1a). However, transmission electron microscopy images show nanoscale needle-like crystals (Figure 1b). 

Individual cellulose nanoparticles and their large aggregates can be found in the aqueous suspension of CNC, and the laser diffraction analysis can determine their size. Since LD-PSA can measure the dimension of spherical particles, this method overestimates the size of non-spherical rod-like cellulose nanocrystals. Transmission electron microscopy, however, can visualize the size and shape of the individual cellulose whiskers on the TEM images, and their length and width can be determined accurately using a special software.

In a previous study, sizing many cellulose nanocrystals with TEM images revealed that the average length of nanocrystals produced from cotton cellulose was 57 nm and the width was 6 nm [20]. TEM images also show that the nanoparticles form large aggregates (Figure 1b), which can be measured in the micrometer range by LD-PSA measurements. Therefore, the aggregation of needle-shaped nanoparticles can be the explanation for LD-PSA results. 

### 3.2. Cellulose Nanocrystals Films

#### 3.2.1. Appearance, Surface Roughness, and Morphology 

In this research, all the films investigated contained 15% glycerol as a plasticizer to reduce the CNC films’ original brittleness. All the films were transparent, colorless, flat, and smooth. First, the effect of heating, citric acid addition, and the combined effect of heating and citric acid were evaluated on the morphology of films. SEM and AFM were used to observe the cryogenic fracture surfaces and to determine the surface roughness, respectively.

As SEM images of the CNC films (Figure 2) show, the inner surface of films presents a closely packed structure with some degree of order in the arrangement of nanocrystals, which resulted in a characteristic layered structure. This parallel alignment of the particles could be explained by preserving the self-assembled cellulose nanocrystals present in the aqueous suspension during film casting and evaporation [21]. While the film made with glycerol only (Figure 2a) has a distinct fibrous character, the nanocrystals in the other films have no well-defined outline and appear to be embedded in a matrix. The formation of the matrix-nanoparticle-like system may be attributed to the heat treatment (Figure 2b), citric acid (Figure 2c), or the combined effect of heat treatment and citric acid (Figure 2d). The citric acid acts as a plasticizer in the former case and as a crosslinking agent in the latter case, combined with the heat treatment, whose processes were examined henceforward.

In order to characterize the surface changes, a more detailed morphological investigation was performed by AFM on the surface of neat film (prepared with 15% glycerol) and CA-30% film (prepared with 15% glycerol and 30% citric acid). Figure 3 shows the typical 3D images with 25 µm^2^ scan areas. The surface roughness of the neat film (R_q_ = 5.3 nm) was significantly higher as compared to the CA-30% film (R_q_ = 2.8 nm). It is assumed that the presence of high concentrations of citric acid, which acts as a plasticizer in the system and fills the voids between the nanocrystals, creates a smoother surface (Figure 3b) by removing the anisotropic contours of the nanocrystal aggregates observed on the surface of the neat film (Figure 3a). Regardless of the significant difference in the roughness of the two films, the absolute values of the roughness seem negligible, revealing the even and smooth surface of both CNC films. Nevertheless, the surface roughness cannot be negligible as it can affect both the interaction with water and the mechanical properties, i.e., the properties that will be investigated later. 

#### 3.2.2. Contact Angle and Surface Free Energy 

For characterizing the surface properties, measuring the contact angle using liquids with significantly different surface tension and the surface free energy that can be calculated from these contact angle values is well-suited. The surface free energy of CNC-based films can provide important information for selecting possible applications. The effectiveness of the methods used to tune the properties of CNC films—in our case, heating, citric acid, and their combined application—can also be indicated by changes in surface energy. Cellulose molecules are hydrophilic, and the same for CNC-based films. For the surface of the films, the contact angle values measured with water and diiodomethane are low and vary in a narrow range of 26–55° (Table 1).

The heat treatment of films prepared without citric acid (0% citric acid concentrations in Table 1) has only a negligible effect on the surface free energy, which can be explained by the migration of some water and glycerol in different proportions at different temperatures. For the samples without heat treatment, the amount of citric acid has no effect on the surface tension (the values are in the range of 68–69 mJ/m^2^), and the measured contact angles are identical within the error range. 

For heat-treated samples up to a CA concentration of 20%, the citric acid has virtually no effect, and neither the contact angle nor the surface tension calculated from it changes. At 30% citric acid concentration, however, the degree of crosslinking can be significant, resulting in a substantial reduction in the number of free and accessible hydroxyl groups on the surface and, hence, a decrease in surface free energy.

As shown in Table 1, the surface tension of CNC films decreases by about 10 mJ/m^2^, from about 65–70 to 54–57 mJ/m^2^, at 30% citric acid concentration and heat treatment. These results are consistent with the previous finding that, when crosslinking occurs in the system, the polar character, and, hence, the surface tension, decreases significantly as OH groups are replaced by less polar ester groups [22]. Similar tendencies and values for surface tension were published for the CNC films crosslinked with an amino-aldehyde compound. It was also confirmed that the amount of crosslinking agent affects the surface energy [4].

The surface groups of the CNC (i.e., surface polarity) determine the surface energy of the nanocellulose and, thus, its wettability. Surface OH groups produce relatively high surface energy (66–70 mJ/m^2^) and good wettability with polar water (i.e., small contact angles: 30–35°). If the amount or nature (ester formation) of the surface OH groups are changed by citric acid, heat treatment, or a combination of both, the surface energy and wettability may also change. The data in Table 1 indicate that neither heat treatment nor citric acid causes a significant change in surface energy and wettability with polar (water) or non-polar (diiodomethane) solvents. 

However, a sufficiently large amount of citric acid (30%) and a simultaneous heat treatment show an apparent change in surface energy and polarity. Irrespective of the annealing temperature applied, the surface energy and polarity decrease. As a result, the wettability by water is reduced, as shown by the significantly increased contact angles (51–54°). A decrease in the polar component of the surface energy causes a reduction in surface energy (~66 → ~55 mJ/m^2^). The dispersion component remains practically unchanged, as shown by the stability of the contact angles measured with the non-polar solvent within the error range. The esterification of OH groups causes a decrease in surface polarity.

#### 3.2.3. Interaction with Water Vapor and Liquid Water

The following experiments investigated the interaction of CNC films with water vapor and liquid water. To measure the water vapor uptake, first, the samples were placed in a desiccator at room temperature at 55% relative humidity for 24 h. Then, they were put into a climate chamber with an RH of 80% at 25 °C and the change in mass of the different films was weighed at fixed intervals until the equilibrium water vapor uptake was reached. We also determined when the disintegration of films laid onto the surface of distilled water started.

The films conditioned at 55% humidity will absorb additional water in the climate chamber, where the relative humidity is 80%. The water vapor uptake of four selected films (neat, neat-heated-160 °C, CA-30%, and CA-30%-heated-160 °C) as a function of time elapsed is shown in Figure 4.

For the neat films (red symbols), a 16% weight increase is measured in 1 h, followed by a gradual decrease in water uptake, reaching a value of less than 1% after 16 days. The significant initial water vapor sorption can break the hydrogen bonds and contribute to the film’s swelling. The swollen open structure can, in turn, promote the migration of glycerol out of the solid film, which can cause significant mass loss. The neat film, containing only glycerol, cannot resist the disintegrating action of liquid water and will break up into larger pieces (Figure 5a) after 10 min.

The water absorption curve of the film heated at 160 °C (blue symbols) is similar to that of the neat film. While the intensive water vapor sorption can explain the initial rapid and significant mass gain, the subsequent decrease is due to glycerol migration from the swollen and loose film structure. It is worth noting, however, that a ratio of glycerol can migrate out of the system during heating at 160 °C. The residual glycerol can leave the solid film in the 80% vapor space. The process is slightly slower, most probably due to the denser structure of the heated film. The heating of the film can also affect the resistance to liquid water. The disintegration is slowing down and takes about 25 min.

For the film containing 15% glycerol and 30% citric acid, the citric acid acts as a plasticizer and interacts not only with the cellulose nanocrystals but also with the glycerol [23]. As a result, glycerol cannot escape from the swollen film and does not cause weight loss at 80% humidity. Similar observations were published for the cellulose–poly(ethylene glycol)–citric acid system, where the esterification prevents PEG from leaching out of the cellulose [24]. Furthermore, all three components—cellulose, glycerol, and citric acid—can bind a significant amount of water vapor (22%) through their hydroxyl groups, which further increases by a few percent in the climate chamber at 80% humidity, as observed via the black curve in Figure 4.

The 15% glycerol and 30% citric acid make the CNC film highly sensitive to liquid water. The disintegration of the loose structure containing a significant amount of plasticizer (15% + 30%) occurs within a few minutes. The nanocrystals are embedded in a high amount of plasticizers as a matrix. The rapid swelling releases the nanocrystals, and, in contrast to the disintegration mechanism of the previously discussed films, the CA-30% film disintegrates into fibrils within a short time (Figure 5b).

It is well-known that esterification of hydroxyl groups of cellulose with citric acid takes place at high temperatures [11,23]. Therefore, in the CNC films discussed previously, no esterification in the CNC/glycerol/citric acid system occurred. For films heated at 160 °C, the esterification and crosslinking are confirmed by the properties of the films. Although the water vapor sorption of free citric acid and cellulose molecules is significant, the formation of the ester group reduces the number of free hydroxyl groups, resulting in a slightly lower water uptake (magenta curve in Figure 4) than that measured for the non-heat-treated CA-30% films (black curve, Figure 4). Furthermore, the crosslinking process results in a fixed film structure, which can be characterized by limited accessibility and swelling. These all explain the resistance of films to the disintegrating effect of liquid water. The heat-treated films are stable and did not degrade over the 48 h study period (Figure 5c).

CNC films crosslinked with an amino-aldehyde compound behaved similarly in water [4]. It was observed that the films with lower crosslinking agent content (0, 2.5, and 5%) disintegrated at higher crosslinker concentrations (10% or more); however, they retained their shape.

In a further experiment, the swelling of these films was characterized. After immersion in distilled water at room temperature for 24 h, the excess water was removed from the surface of films and the mass was measured. The swelling rate of films depends significantly on the temperature of heating treatment and citric acid concentration (Table 2). No crosslinking occurs during heat treatment at 100 °C; independently from the citric acid concentration, the films delaminate in a short time by a progressive and infinite swelling. Thus, for the non-crosslinked films, the swelling rate is not possible to measure.

A minimum heating temperature of 120 °C and a citric acid concentration of 20% are required to obtain a stable CNC film structure resistant to liquid water. The crosslinked films retain their shape and, besides the significant swelling, their weight can be measured. The results also reveal that increasing the concentration of crosslinking agent (from 10 to 30%) and the curing temperature (from 120 to 160 °C) continuously decreases the swelling rate of films (from 250% to 110%). The decreasing swelling values indicate the increasing crosslinking density in the films. 

Heat-assisted ester bond formation and crosslinking in cellulose nanofibers with citric acid was confirmed by ^13^C NMR spectroscopy [7], and the reduced swelling of substrates was explained by the higher crosslinking density. In several studies, FT-IR spectroscopy was used to confirm the interaction between cellulose/starch and citric acid and the ester bond formation [10,15,25]. The lower water uptake of regenerated cellulose films was explained with crosslinking [15]. A reduction in moisture content and moisture uptake was also observed in starch-based films due to the crosslinking with citric acid [9]. The authors found that the bonds formed with citric acid supplement the intermolecular hydrogen bonds naturally present in starch, improving water resistance.

#### 3.2.4. Mechanical Properties of the CNC Nanocomposite Films

It is well-known that cellulose nanocrystal films have excellent mechanical properties [26]. The values determined with the tensile test highly depend on the quality and quantity of additives added to the CNC, such as plasticizers and crosslinking agents [27,28]. Our results confirm this finding. The tensile properties of the neat and CNC films prepared with CA crosslinker in a wide concentration range, without (Figure 6) and with (Figure 7) a heating treatment at 160 °C for 10 min, were tested. The neat film, cast from cotton-CNC suspension with only 15% glycerol plasticizer, has a high tensile strength (about 25 MPa) but a low elongation of only around 1%, as indicated by the symbols on the *Y* axis.

As mentioned before, without heat treatment, citric acid acts as a plasticizer in CNC films, affecting many film properties, including mechanical properties. Increasing the concentration of citric acid to 30% significantly decreases the tensile strength of the CNC films (from about 25 MPa to 15 MPa, Figure 6a) and increases the deformability (elongation at break) from about 1% to 5% (Figure 6b). In summary, flexible, transparent, and easy-to-handle films are concluded when 15% glycerol and 20–30% citric acid are added, as illustrated in the photo inset in Figure 6b.

Applying heat treatment and citric acid can lead to ester formation and crosslinking of the hydroxyl groups available in the cellulose nanocrystals. The mechanical properties depend on the ratio of crosslinked to free citric acid molecules, which, in the latter case, act as a plasticizer. The crosslinked citric acid increases the strength and reduces the deformation (Figure 7), whereas the free citric acid has the opposite effect on the tensile strength and elongation at break, as already shown in Figure 6. 

Similar effects were observed in regenerated cellulose film crosslinked with citric acid [15]. Increased tensile strength was also measured for citric acid–starch–cellulose foams up to a citric acid concentration of 5%. Above 5%, however, a decrease in tensile strength was observed and explained by the citric-acid-catalyzed hydrolytic degradation of polysaccharides [25]. Citric acid was proven to be an efficient crosslinking agent for nanocellulose to improve the wet strength of the paper filter [10]. 

As shown in Figure 7a, an increase in the concentration of citric acid results in an increase in strength, indicating the formation of crosslinks and, thus, the predominance of chemically bonded citric acid relative to free molecules. It should be noted that the higher strength of the neat film (without citric acid and heating, symbols on the *Y* axis) is explained by the absence of heat treatment and the low plasticizer concentration, which is only 15% glycerol.

There is no increase in elongation at break, only at 30% citric acid concentration (Figure 7b), which can be explained by many crosslinks formed during the heating, fixing the structure and limiting its ability to deform. At the same concentrations of citric acid, the elongation at break of heated samples is lower than that of the non-heated ones, as the results in Figure 6 and Figure 7 reveal.

## 4. Conclusions

It is well-known that cellulose nanocrystals have unique strengthening effects and optical properties, which can be used in nanocomposites, paper making, packaging, gas barriers, etc. [1]. The thin films prepared from cellulose nanocrystals also have excellent properties, such as high transparency, tensile strength, and good barrier properties. During the drying of films, strong hydrogen bonds are formed between the cellulose nanocrystals, resulting in a compact layered structure. However, the resulting interior structure can be disintegrated by water and broken down into nanocrystals. Thus, the resistance of CNC film to water needs to be improved. One of the best ways to produce water-resistant CNC films is the crosslinking of cellulose. 

In this research work, the structure and interactions in nanocrystalline cellulose films were modified using citric acid as a plasticizer and/or crosslinking agent and applying thermal treatment. Thin films were prepared from cellulose nanocrystals with glycerol plasticizer at a 15% concentration by casting and evaporation. Citric acid was added to the CNC in different percentages (0, 10, 20, and 30%), and the thermochemical reaction of citric acid and cellulose was investigated by applying short (10 min) heating at different temperatures (100, 140, and 160 °C).

While the heat treatment alone did not significantly change the film structure and properties, the amount of citric acid and the combined effect of citric acid and heat treatment significantly modified the film properties. The results confirmed that citric acid is a plasticizer in CNC films without heat treatment, resulting in an easy-to-handle and flexible film structure. Citric acid with heat treatment, however, results in a crosslinked structure, making the films stiffer but with excellent tensile strength and resistance to water. A reduction in surface energy (~66 → ~55 mJ/m^2^) and wettability, characterized by increased water contact angles (from ~30° to ~50°), also confirms the crosslinked structure. Stable water-resistant films can be achieved with 20% citric acid concentration and short heating above 120 °C. The water resistance can open new areas for the application of CNC films.

## Figures and Tables

**Figure 1 polymers-15-01698-f001:**
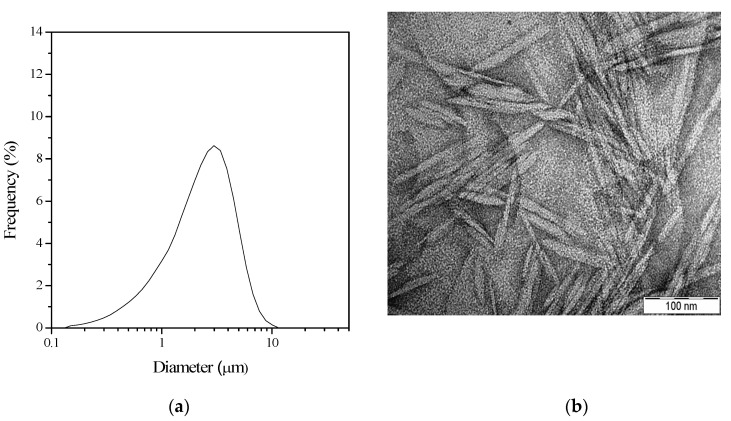
(**a**) Particle size distribution of CNC in an aqueous suspension determined by laser diffraction particle size analysis; (**b**) TEM micrographs (18,000×) of CNC particles in aqueous suspension.

**Figure 2 polymers-15-01698-f002:**
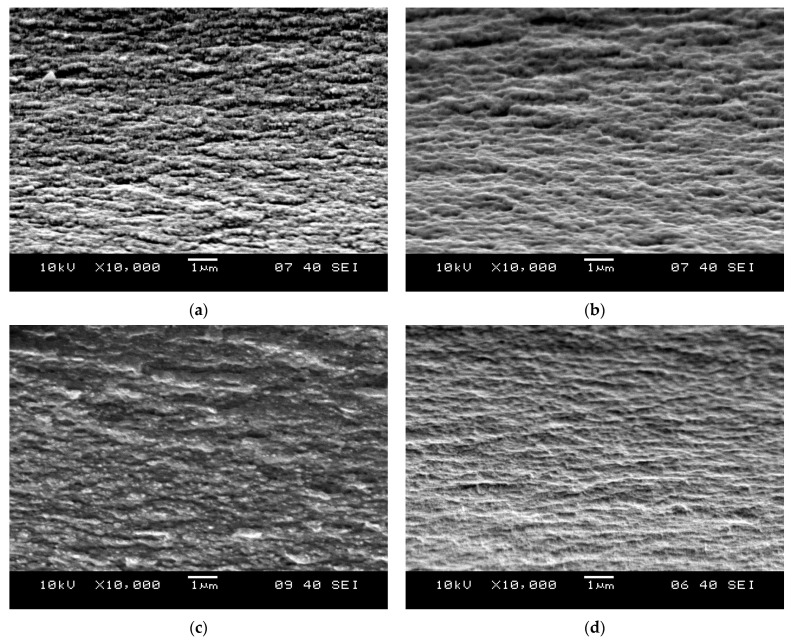
SEM images of cryogenic fracture surfaces of CNC films plasticized with 15% glycerol; (**a**) neat; (**b**) neat after heating at 160 °C for 10 min; (**c**) CA-30% film; (**d**) CA-30% film after heating at 160 °C for 10 min (M: 10,000×).

**Figure 3 polymers-15-01698-f003:**
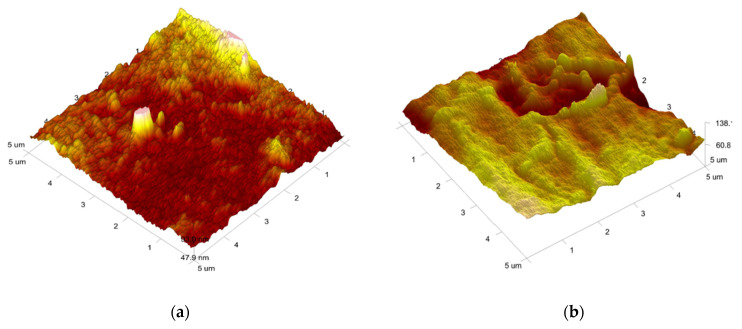
AFM images of the surface morphology of CNC films plasticized with 15% glycerol; (**a**) neat, (**b**) CA-30%. Roughness parameter: neat: R_q_ = 5.3 nm; CA-30%: R_q_ = 2.8 nm.

**Figure 4 polymers-15-01698-f004:**
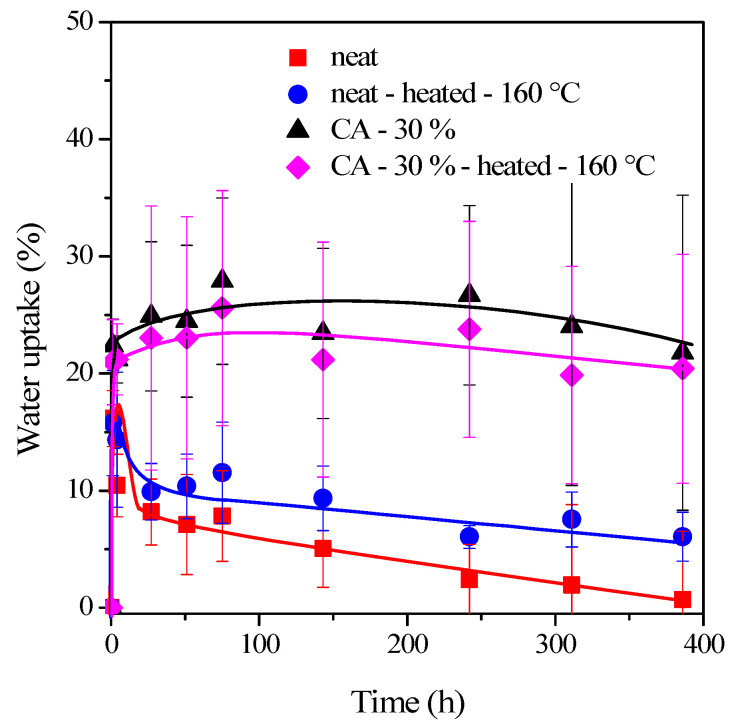
Water vapor uptake of the CNC films (plasticized with 15% glycerol and conditioned at 55% RH and room temperature) in a climate chamber with an RH of 80% at 25 °C as a function of time. The effect of heating of films at 160 °C for 10 min and citric acid addition at 30% concentration.

**Figure 5 polymers-15-01698-f005:**
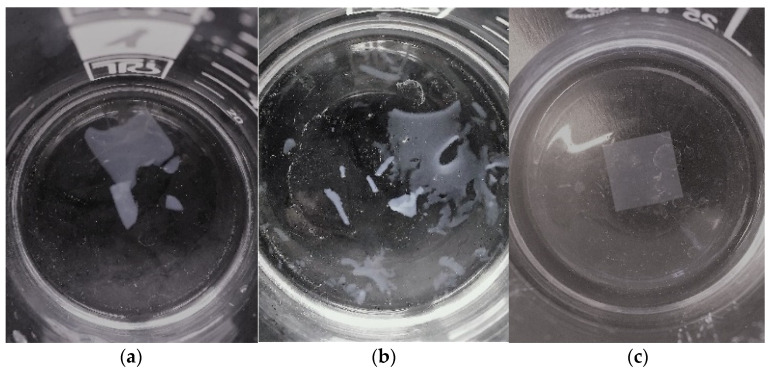
Interaction of CNC films, plasticized with 15% glycerol, with liquid water. Photos were taken after 10 min treatment. (**a**) Neat film; (**b**) CA-30% film; (**c**) CA-30% film heated at 160 °C for 10 min.

**Figure 6 polymers-15-01698-f006:**
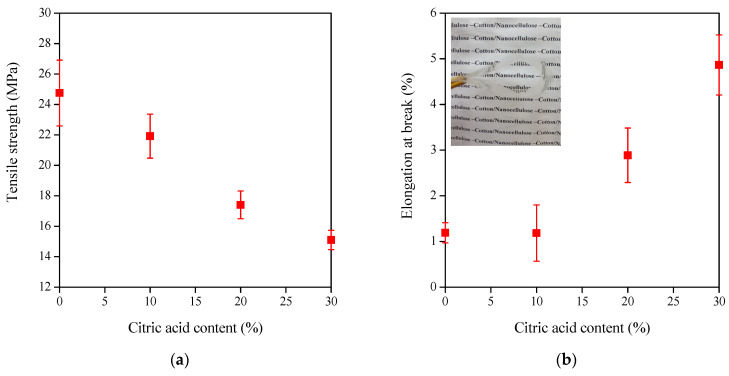
Tensile strength (**a**) and elongation at break (**b**) of CNC-CA films plasticized with 15% glycerol as a function of citric acid concentration. Inset in Figure (**b**) represents the photo of the flexible CA-30% film.

**Figure 7 polymers-15-01698-f007:**
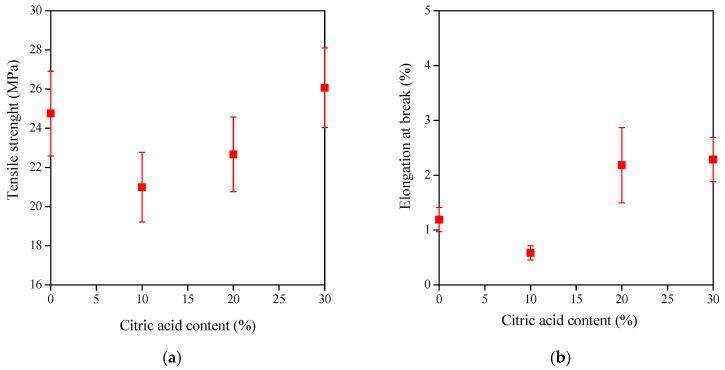
Tensile strength (**a**) and elongation at break (**b**) of CNC-CA films plasticized with 15% glycerol and heated at 160 °C for 10 min as a function of citric acid concentration.

**Table 1 polymers-15-01698-t001:** The effect of heating at 100, 120, and 160 °C for 10 min, citric acid addition at 0, 10, 20, and 30% concentrations on the contact angle measured with water and diiodomethane, and surface tension of CNC films plasticized with 15% glycerol.

Heating (°C)	Citric Acid Concentration (%)	Contact Angle (°)		Surface Tension (mJ/m^2^)
		Water	Diiodomethane	
-	0	35.2 ± 1.9	38.5 ± 6.1	66.5
	10	32.3 ± 5.6	33.4 ± 5.3	69.1
	20	31.9 ± 4.3	35.6 ± 4.8	68.3
	30	31.4 ± 2.6	38.0 ± 1.7	68.5
100	0	27.2 ± 2.3	36.4 ± 7.2	70.8
	10	36.6 ± 4.7	26.8 ± 6.8	68.3
	20	31.1 ± 3.1	35.4 ± 2.3	69.2
	30	54.0 ± 1.2	43.2 ± 1.6	54.4
120	0	36.4 ± 0.8	36.9 ± 6.5	66.2
	10	38.8 ± 1.4	34.7 ± 1.7	65.5
	20	32.2 ± 2.3	37.2 ± 1.8	68.3
	30	51.1 ± 1.5	38.1 ± 2.9	57.7
160	0	38.7 ± 1.4	37.2 ± 9.1	64.9
	10	42.3 ± 6.5	36.5 ± 1.6	63.1
	20	37.6 ± 4.2	38.4 ± 2.1	65.2
	30	54.2 ± 3.8	39.4 ± 3.8	55.5

**Table 2 polymers-15-01698-t002:** Swelling rate of CNC films plasticized with 15% glycerol and determined after immersion in water for 24 h. The effect of heating at 100, 120, and 160 °C for 10 min, and citric acid addition at 10, 20, and 30% concentrations.

Heating (°C)	Citric Acid Concentration (%)	SR (%)
	10	d ^1^
100	20	d ^1^
	30	d ^1^
	10	d ^2^
120	20	250 ± 15
	30	150 ± 10
	10	160 ± 10
160	20	150 ± 10
	30	110 ± 5

^1^ Disintegration in 10–32 min. ^2^ Disintegration in 50 min.

## Data Availability

The data presented in this study are available on request from the corresponding author.

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
