# Peer review of "Effect of Heating and Citric Acid on the Performance of Cellulose Nanocrystal Thin Films"

_polymers, 2023, doi:10.3390/polym15071698_

Round 1

Reviewer 1 Report

Paper title: Effect of heating and citric acid on the performance of cellulose 2 nanocrystal thin films

Introduction

A better statement of what was the rationale and how important the subject addressed in the research. It seems that the authors did not know how to highlight the importance of their own work that they report.

Section 2.3. Casting of CNC-films

Lines 104-107 The paragraph is not clear... need to be rewritten.

Line 108: "as described above." but there is no description!!

Results

Line 166: The aqueous CNC suspensions contained about 3 % cellulose nanocrystals by weight ??? Which suspensions? From where??

Can authors explain why there are so different sizes of CNC acquired by DLC and TEM ?

To have a better insight into the interactions that occurred FTIR spectra of the seected samples should be presented.

Conclusions sections

The following paragraphs are useless and general, without saying anything about the outcomes of the work:

"Thin films prepared from cellulose nanocrystal suspensions by casting and evaporation have excellent and unique properties. However, their interaction with water needs to be improved."?!

Reviewer 2 Report

The manuscript of Emília Csiszár et al "Effect of heating and citric acid on the performance of cellulose nanocrystal thin films" is devoted to obtaining films based on cotton nanocellulose and modifying them with glycerol and citric acid. In my opinion, the literature review in the manuscript is very brief and should be expanded. For the pulp used, it is necessary to provide characteristics (SP, alpha content, etc.). It is important to compare the obtained results with those already presented in the literature. The novelty of the work should be more clearly presented. It is not clear what happens to the structure of the films. In my opinion, it is necessary to bring the RSA data. This will help to understand the presented data regarding the mechanics and sorption properties. How do the authors explain the high sorption of CA-30% samples (heated 160°C) and the low elongation values?

Line 17. "liquid" - I recommend deleting.
2.1. materials. The choice of materials for work is not entirely clear. Why didn't the authors use original cotton?
Line 85. "CNC-s" should be replaced with "CNC"
Line 189. I propose replacing "neat film" with "film"
Table 1. From the given data in the table, it is not clear the influence of temperature and acid on Surface tension and Contact angle. Authors should clearly articulate the reasons for changing these parameters.

Round 2

Reviewer 1 Report

The revised version can be accepted.

Author Response

Thank you for the acceptance of our manuscript. 

Reviewer 2 Report

"The alpha-cellulose content of bleached cotton is 100%." - I do not quite agree with the authors, after processing, cotton may contain fats, resins. This, in turn, will affect the properties of the system. "It is not clear what happens to the structure of the films. In my opinion, it is necessary to bring the RSA data. This will help to understand the presented data regarding the mechanics and sorption properties." - structural data obtained by X-ray (it is desirable to provide diffractograms). "The surface energy, wettability, and reduced disintegration in the water suggest that in the heat-treated CA-30% samples, the citric acid molecules crosslink the CNC particles, resulting in a reduction of the film deformability (elongation in Figure 7b). However, the amount of water bound to the sample is not significantly reduced (Figure 4, symbols: magenta). the value measured for all other samples at the beginning of the test (all symbols on Y axes at 0 time)." - in my opinion, it is the diffraction patterns and the data obtained from them (the degree of crystallinity and the size of the crystallites) that will make it possible to understand this phenomenon. At the moment, I do not understand the answer presented by the authors and doubt the correctness of the data presented in the manuscript.

L. 248, 249. I recommend rewriting this statement.

The quality of Figure 4 is desirable to improve.

Author Response

March 22, 2023

Answers

to the comments of the Reviewer_2 on the manuscript “Effect of heating and citric acid on the performance of cellulose nanocrystal thin films” by Csiszár, E., Herceg, I., and Fekete, E. submitted to the Polymers (Manuscript ID: polymers-2283507)

"The alpha-cellulose content of bleached cotton is 100%." - I do not quite agree with the authors, after processing, cotton may contain fats, resins. This, in turn, will affect the properties of the system. 

Reply:

After alkaline scouring and bleaching, the most important processes in cotton preparation technology, (bleached) cotton contains only cellulose, and the traces of incrusting materials (if any) do not affect the subsequently applied sulphuric acid hydrolysis with a concentration of 64 %.

"It is not clear what happens to the structure of the films. In my opinion, it is necessary to bring the RSA data. This will help to understand the presented data regarding the mechanics and sorption properties." - structural data obtained by X-ray (it is desirable to provide diffractograms). 

"The surface energy, wettability, and reduced disintegration in the water suggest that in the heat-treated CA-30% samples, the citric acid molecules crosslink the CNC particles, resulting in a reduction of the film deformability (elongation in Figure 7b). However, the amount of water bound to the sample is not significantly reduced (Figure 4, symbols: magenta). the value measured for all other samples at the beginning of the test (all symbols on Y axes at 0 time)." - in my opinion, it is the diffraction patterns and the data obtained from them (the degree of crystallinity and the size of the crystallites) that will make it possible to understand this phenomenon. 

At the moment, I do not understand the answer presented by the authors and doubt the correctness of the data presented in the manuscript.

Reply:

Unfortunately, we do not understand exactly what the reviewer suggested (RSA data?). What does ‘RSA’ mean? What does this acronym ‘RSA’ mean?

Unfortunately, in the frame of the current project, we cannot perform an additional characterization with any methods. Nevertheless, we refuse to question the “correctness” of the data. Numerous investigations, measurements and experience in this research field support our findings.

  1. 248, 249. I recommend rewriting this statement.

Reply:

The statement has been rewritten.

The quality of Figure 4 is desirable to improve.

Reply

The quality of Figure 4 has been improved.